# New Datasets and Controllable Iterative Data Augmentation Method for Code-switching ASR Error Correction

**Zhaohong Wan**[1,2] and **Xiaojun Wan**[1,2] and **Wei Peng**[3] and **Rongjun Li** [3]

[1]Wangxuan Institute of Computer Technology, Peking University
[2]The MOE Key Laboratory of Computational Linguistics, Peking University
[3] Artificial Intelligence Application Research Center, Huawei Technologies
{xmwzh,wanxiaojun}@pku.edu.cn

## Abstract

With the wide use of automatic speech recognition(ASR) systems, researchers pay more attention to the ASR error correction task to improve the quality of recognition results. In particular, ASR in bilingual or multilingual settings, namely code-switching ASR, has greater challenges and research value. In this paper, we first present code-switching ASR correction datasets obtained from solid ASR systems and automatic annotators. The datasets contain Chinese-English code-switching dialogues of bilingual speakers in Singapore, Malaysia, and Hong Kong. Based on this task, we propose a controllable iterative (CI) data augmentation method for improving the performance of mainstream ASR error correction systems. With a small amount of training data, our proposed method has the ability to iteratively produce abundant pseudo parallel data from the monolingual corpus for Chinese-English code-switching ASR correction. Results of experiments show that our method achieves the best performance compared with the rule-based, back-translation-based data augmentation methods and large language model Chat-GPT.

## 1 Introduction

Recently, automatic speech recognition (ASR) system has achieved impressive success with the adoption of advanced deep neural network (Moritz et al., 2020). Learning from training data, the ASR system can match human transcribers for some critical tasks. It has been widely applied in several areas, such as live captioning, intelligent appliance control (Aliprandi et al., 2014).

In some bilingual and multilingual societies, people often speak more than one language in a conversation. The speech phenomenon of switching language among sentences is called code-switching. To explore the code-switching phenomenon, code-switching tasks in many different

| Error Type | Redundant |
|---|---|
| Source | 我 要 start on 我的 a essay
(I want to start on my a essay.) |
| Target | 我 要 start on 我的 essay
(I want to start on my essay.) |

Table 1: Examples of Chinese-English code-switching ASR error correction in our datasets. The contents in brackets are the translations for non-native Chinese speakers.

language pairs have been introduced (Dowlagar and Mamidi, 2023; Seddah et al., 2020), including the code-switching ASR task.

Employing multilingual ASR systems to solve code-switching ASR is proposed in recent work (Lovenia et al., 2021). However, the performance of systems in many cases is still far from satisfactory. Errors introduced by ASR systems usually affect the performance of downstream tasks. To improve the recognition performance, the code-switching ASR error correction task is necessary.

In this paper, we focus on code-switching ASR error correction between Chinese and English, which is common in the Chinese communities of Asia area. To enable the research and development of code-switching ASR error correction, we propose two datasets SEAME-C and ASCEND-C that are constructed from two Chinese-English code-switching ASR datasets SEAME (Lyu et al., 2010) and ASCEND (Lovenia et al., 2021). In Table 1, we present typical examples of Chinese-English code-switching ASR error correction.

A challenge for the code-switching ASR error correction is the lack of a large scale of training data. Moreover, we can not even collect sufficient high-quality Chinese-English code-switching data. Considering these challenges, we propose a controllable iterative (CI) data augmentation method that can iteratively generate abundant code-switching ASR error correction instances from the monolingual corpus with small-scale labeled training data.

To verify the effectiveness of our proposed method, we implement two data augmentation baselines and evaluate the performance of these methods on proposed datasets at first. With a similar scale of augmented data, our proposed controllable iterative method achieves the best performance in both MaxMatch ($M^2$) scorer (Dahlmeier and Ng, 2012) and MER metrics on SEAME-C and ASCEND-C datasets. It performs well for both mT5 (Xue et al., 2021) and mBART (Tang et al., 2020) correction methods. Besides, considering the strong ability of large language models (LLM), we compare the performance of our method with the most powerful LLM ChatGPT. Experiments show that our method outdistances the ChatGPT method, which shows the effectiveness of our method in the Chinese-English code-switching ASR error correction task. Besides, we find that this task is challenging, and LLM method is far from achieving satisfactory results at now.

Our contributions are summarized as follows:

1. We propose two datasets SEAME-C and ASCEND-C for the challenging Chinese-English code-switching ASR error correction task.

2. To address the problem of lacking sufficient training data, we propose the controllable iterative data augmentation method that can iteratively generate abundant code-switching ASR error correction instances from the monolingual corpus.

3. Extensive experiments show the superiority of our proposed controllable iterative method. Moreover, combining the pseudo data produced by the rule-based and controllable iterative methods can further improve the performance of error correction models.

The resources are publicly available[1].

## 2 Related Work

Code-switching ASR, the task aiming to recognize the speech that contains more than one language within the sentences, has been increasingly reported in speech technology and linguistic studies recently (Su, 2001; Auer, 2013). However, to our knowledge, there is no research on the code-switching ASR error correction task at present. In this section, we introduce two relevant fields: ASR error correction and data augmentation method in text error correction.

With the successes of ASR systems, the ASR error correction has become a prevalent task in recent years. The idea of regarding the ASR error correction as the machine translation and applied advanced neural machine translation models is adopted in several previous approaches(Mani et al., 2020; Wang et al., 2020). D'Haro and Banchs (2016) first introduce machine translation to improve automatic transcription. Considering the machine translation models require only parallel text pairs, Guo et al. (2019) propose an approach to utilizing text-only data by training a correction model to explicitly correct the errors. The joint modeling method shows great performance in this task(Wang et al., 2023). Re-ranking ASR n-best hypotheses is also a popular topic for error recovery (Corona et al., 2017; Jonson, 2006).

To address the problem of lack of training data, data augmentation method is widely adopted in text error correction area. Many works (Grundkiewicz et al., 2019; Lichtarge et al., 2019) adopt pre-defined rules to generate sentences with errors. Inspired by back-translation procedure for machine translation (Sennrich et al., 2016), works (Xie et al., 2018; Kiyono et al., 2019)are proposed to train a model to generate erroneous sentences. Recently, researchers focus on the exposure bias problem of data augmentation method and proposed methods to utilize the augmented data in more effective ways (Solyman et al., 2023; Cao et al., 2023).

## 3 Code-switching ASR Error Correction Datasets

In this section, we provide details about our two datasets SEAME-C and ASCEND-C. Specifically, we give the construction process of our datasets at first. Then, we do an analysis of our datasets and show some characteristics of them. In this work, we adopt the same corpus splitting as the original ASR datasets. In addition, we remove some bad cases that are too short and eliminate monolingual sentences in the validation and test sets to ensure the quality of evaluation. Our datasets provide quality resources for the research and development of code-switching ASR error correction. Table 2 describes the statistics of our proposed datasets.

### 3.1 Data Construction

In some Asia's multilingual countries and regions, speaking a mixture of Chinese and English within a sentence is common. We collect data from two Chinese-English code-switching ASR datasets with

---

[1]https://github.com/chopper2k/Code-switching-ASR-Error-Correction

|  | SEAME-C | | | ASCEND-C | | | |
|---|---|---|---|---|---|---|---|
|  | Train | Test | Total | Train | Val | Test | Total |
| # Dialogues | 267 | 30 | 297 | 40 | 5 | 4 | 49 |
| # Sentences | 48,405 | 6,293 | 54,698 | 9,869 | 213 | 373 | 10,455 |
| # Tokens | 801,548 | 99,837 | 901,385 | 122,348 | 3,385 | 5,538 | 131,271 |
| Avg. sentences in a dialogue | 181.3 | 209.8 | 174.2 | 246.7 | 53.3 | 92.3 | 213.4 |
| Avg. tokens in a sentence | 16.6 | 15.9 | 16.5 | 12.4 | 15.9 | 14.8 | 12.6 |

Table 2: Statistics of our proposed code-switching ASR error correction datasets.

the advanced ASR system and automatic annotator. In this part, we introduce the main components of the data construction. Details of processing and implementation are shown in the Appendix A.

**ASR Corpus**     We build the error correction instances from two Chinese-English code-switching ASR datasets: SEAME and ASCEND. SEAME is a large Chinese-English code-switching spontaneous speech corpus, collected from residents of Malaysia and Singapore. The corpus contains 30 hours of word-level transcribed code-switching speech data. ASCEND is a high-quality Chinese-English code-switching corpus built on spontaneous multi-turn conversation collected in Hong Kong. It comprises 10.62 hours of clean speech data collected from dialogues.

**ASR System**     With code-switching ASR datasets, we adopt the ASR system to obtain recognition results. For being close to the real recognition setups, we employ an advanced ASR model architecture, namely XLSR-53 (Xu et al., 2021). We fine-tune the XLSR-53 model and obtain the recognition results for datasets. The code-switching ASR error correction instances in our datasets are built by the pairs of manually annotated transcriptions and corresponding recognition results.

**Annotator**     We develop an automatic annotator for the Chinese-English code-switching corpus, based on prior works in English and Chinese (Bryant et al., 2017; Hinson et al., 2020). It takes three steps, tokenization, alignment, and merging, to generate annotations that are suitable for Max-Match ($M_2$) evaluation metrics. Besides, the automatic annotator is able to classify errors into four types: redundant (R), missing (M), word selection (S), and word ordering (W). It allows researchers to conduct more detailed experiments and analyses with different error types.

### 3.2 Analysis

#### 3.2.1 Language Distribution

As the code-switching datasets contain multiple languages, the observation of language distribu-

|  | SEAME-C | ASCEND-C |
|---|---|---|
| **At dialogue level** | | |
| Code-switching dialogues | 100% | 100% |
| **At sentence level** | | |
| Code-switching sentences | 76.6% | 27.8% |
| Chinese sentences | 14.3% | 48.6% |
| English sentences | 9.1% | 23.6% |
| **At token level** | | |
| Chinese tokens | 83.7% | 92.5% |
| English tokens | 16.3% | 7.5% |

Table 3: Proportion of languages used in datasets at dialogue, sentence and token levels.

tion is important for the analysis. We calculate the proportion of languages used in the datasets at dialogue, utterance, and token levels. Table 3 presents the details.

From the table, we can find that the sentence-level language distribution of two datasets is quite different. In the SEAME-C dataset, the code-switching sentences take the main proportion, accounting for 76.6%. As for the ASCEND-C dataset, the Chinese sentences account for the largest proportion, while the code-switching instances only account for 27.8%. In terms of token-level language distribution, both two datasets are dominated by Chinese tokens. It shows that the code-switching phenomenon in Asia area mainly occurs by mixing a few English words into Chinese speech. Considering that, we generate pseudo data from Chinese sentences in our data augmentation method in the following sections.

#### 3.2.2 Error Distribution

With the proposed automatic annotator, the errors in datasets are classified into four types. To analyze the characteristic of ASR errors in proposed datasets, we calculate the error distributions on the test sets of SEAME-C and ASCEND-C datasets. The results are presented in Table 4.

As shown in the table, the error distributions on the test sets of two datasets are similar. The word selection errors account for the largest proportion, with 85.7% in SEAME-C and 81.3% in ASCEND-C. This type of error is caused by mistakes in the

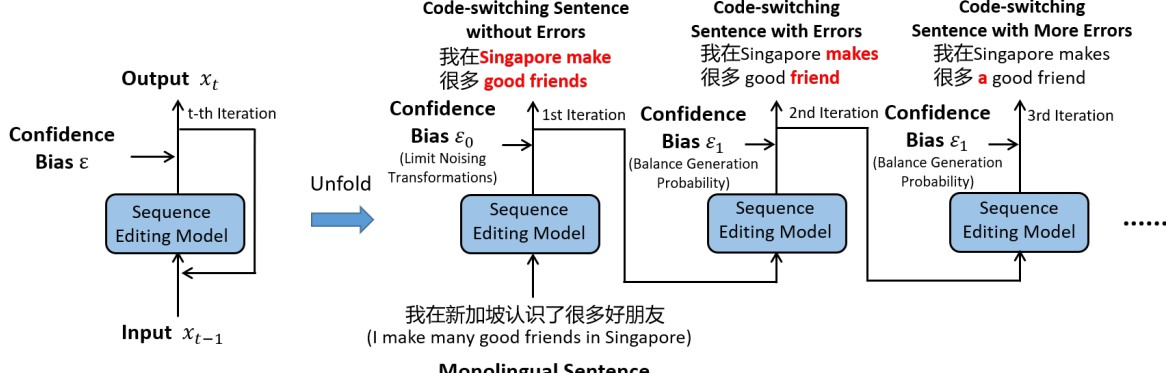

Figure 1: An example of the process of generating pseudo data in our controllable iterative method. We highlight the edited parts of sentences in each iteration. The controllable iterative method produces pseudo training instances by iteratively generating outputs from input sentences. We apply the sequence editing model to generate synthetic data and build training pairs with iterations.

| Error Type | SEAME-C | | ASCEND-C | |
|---|---|---|---|---|
| | # | % | # | % |
| Redundant (R) | 908 | 5.6 | 134 | 11.2 |
| Missing (M) | 1,381 | 8.6 | 86 | 7.2 |
| Word Selection (S) | 13,831 | 85.7 | 972 | 81.3 |
| Word Ordering (W) | 15 | 0.1 | 4 | 0.3 |

Table 4: Error distribution on the test sets of SEAME-C and ASCEND-C datasets. The # columns present the amounts of specific error.

recognition of words or phrases with similar pronunciation. Besides, there are a number of missing and redundant errors, which is challenging to correct for requiring to change the length of the input. Additionally, the frequency of word ordering errors is very few. We can find that the code-switching ASR error correction is a task with imbalanced error distribution, and we need to consider it when designing the correction method.

## 4 Controllable Iterative Data Augmentation Method

To address the problem of lacking sufficient training data, we propose the controllable iterative data augmentation method. Due to the shortage of high-quality Chinese-English code-switching corpus, our method is designed to directly generate training instances from monolingual texts. Our method consists of two parts: sequence editing model and pseudo data generation algorithm.

Sequence editing model is the fundamental structure of our method. It adopts the Transformer architecture and the predefined editing operations called transformations to generate raw augmented data from seed corpus. Pseudo data generation algo-

rithm plays a crucial role in the proposed method. It adopts the sequence editing model to iteratively produce pseudo training data under the control of confidence bias. With our designed transformations, the sequence editing model is able to directly generate training instances from monolingual texts. Figure 1 shows an example of the process of generating pseudo data in our controllable iterative method.

### 4.1 Sequence Editing Model

The data augmentation method for the code-switching ASR error correction task aims to generate instances with errors as pseudo training data. Error generation can be treated as the process of noising the correct source input with edit operations. Based on this idea, we adopt the sequence editing model to obtain the editing sequence required to produce output with errors.

#### 4.1.1 Predefined Transformations

The sequence editing method requires predefined editing operations called transformations. In this work, we adopt the custom token-level transformations developed in GECToR (Omelianchuk et al., 2020), the details are shown in Appendix B. These transformations are originally designed for English texts. We denote these transformations except $TRANSLATION$ as noising transformations. For generating code-switching sentences from monolingual sources, we introduce cross-lingual transformations into original ones. Firstly, we add the translation transformation ($TRANSLATION$) that replaces the current token with the most common translation in a Chinese-English

dictionary. Secondly, to cover the specific cases that the most common translation can not fit, we insert additional Chinese characters to the vocabulary of $APPEND$ and $REPLACE$ operations. With predefined transformations, we can get the target sentence by applying the sequential outputs of transformations to the tokens in the source sentence.

### 4.1.2 Model Architecture

The sequence editing model is an encoder made up of Transformer architecture stacked with two linear layers with softmax layers. Considering the strong performance of pre-trained model, the Transformer architecture is initiated with multilingual pre-trained model m-BERT (Devlin et al., 2019). The model is able to generate an editing operation sequence from the source input, and then apply the operations on the source to obtain the target output.

To make the generation process of the model controllable, we add a confidence bias to the decoding process. The confidence bias is a parameter applied to change the generation probability of transformations. For example, adding a positive confidence bias to $KEEP$ can increase the probability of generating the $KEEP$. By adjusting the confidence bias, the decoding process of sequence editing model is under control.

### 4.1.3 Training Process

To generate pseudo data, we need to train the sequence editing model to generate noisy sentences from correct ones at first.

Given a pair of a monolingual sentence $x$ without noise and a code-switching sentence with noise $y$. Then we convert the sentence $y$ to a sequence of transformations $z$. Let $\epsilon$ denote the preset confidence bias and $\Theta$ denote all trainable parameters of the sequence editing model $\phi$. Our objective is to find the optimal parameter set $\hat{\Theta}$ that minimizes the following negative log-likelihood function:

$$L = -\log P(z|x, \epsilon, \Theta) \tag{1}$$

where $P(z|x, \epsilon, \Theta)$ denotes the conditional probability.

After training on constructed instances, our model can not only learn the editing operations to apply errors to sentences but also the transformations to generate English-Chinese code-switching texts. Some details of training are shown in Appendix C.

---

**Algorithm 1** Generating pseudo data

**Input:** Monolingual Sentence $x_0$; Number of iterations $T$; Sequence editing model $\phi$; Confidence bias $\epsilon$; Similarity Discriminator $\psi$; Hyper parameter $\lambda, \epsilon_0, \epsilon_1$
**Output:** Synthetic erroneous samples set $S$
**Function:** $Gen(x_0, T, \lambda, \epsilon_0, \epsilon_1)$:

1: **for** $t \leftarrow 1$ to $T$ **do**
2:     **if** $t = 1$ **then**
3:         $\epsilon \leftarrow \epsilon_0$
4:     **else**
5:         $\epsilon \leftarrow \epsilon_1$
6:     $x_t \leftarrow \phi(x_{t-1}, \epsilon)$
7:     **if** $t \geq 2$ **then**
8:         $p \leftarrow \psi(x_1, x_t)$
9:         **if** $p \geq \lambda$ **then**
10:           $S.add(x_1, x_t)$
    **return** S

---

### 4.2 Pseudo Data Generation Algorithm

To generate pseudo training data from the monolingual corpus for code-switching ASR error correction, we have to generate both code-switching target sentences and source sentences with ASR errors. For this objective, we propose an iterative generation approach to obtain pseudo training pairs for code-switching ASR error correction with only one model in multiple iterations. The algorithm for generating pseudo training data is summarized in Algorithm 1.

Given a monolingual sentence from seed corpus $x_0$. We apply $T$ times of iterations to the sequence editing model $\phi$ to generate data.

In the first iteration, we adjust the confidence bias to $\epsilon_0$ that limits the noising transformations and makes the model use cross-lingual transformations and $KEEP$ to generate code-switching sentence $x_1$. This process can be formulated as:

$$x_1 \leftarrow \phi(x_0, \epsilon_0) \tag{2}$$

In the $t$-th ($1 < t \leq T$) iteration, the confidence bias is set to $\epsilon_1$ that balances the generation probability of each transformation. It makes the model apply diverse noises to the output of the $(t$-1)-th iteration $x_{t-1}$ and obtain the pseudo instances with errors $x_t$. This process can be formulated as:

$$x_t \leftarrow \phi(x_{t-1}, \epsilon_1) \tag{3}$$

where $1 < t \leq T$.

With above pseudo instances $x_1, \ldots, x_T$, we treat the output in the first iteration as the correct target sentence and the output in the later turn as noised source sentence to get the sentence pairs.

In order to improve the quality of synthetic samples, we use BERTScore (Zhang et al., 2019) as the similarity discriminator to filter sentence pairs with low similarity. Given a sentence pair $(x_1, x_t)$ $(1 < t \leq T)$, we use similarity discriminator $\psi$ to get a score $p \in [0, 1]$ that presents the semantic similarity between sentences $x_1$ and $x_t$. We set a threshold $\lambda$ that if the score $p$ of a sentence pair is greater than the threshold, the pair can be selected as an augmented training sample.

## 5 Experimental Setup

In this section, we present our experimental setup including datasets, ASR error correction model, and data augmentation baseline. More details of implementation are presented in Appendix D.

### 5.1 Dataset for Experiment

**Training and Test Data**    Firstly, we train the model on pseudo training instances generated from data augmentation method. Then, we fine-tune the code-switching ASR error correction model on our proposed Chinese-English code-switching ASR error correction datasets: SEAME-C and ASCEND-C. We take $M^2$ scorer and mixed error rate (MER)(Adel et al., 2013) as evaluation metrics.

**Seed Corpus for Pseudo Training Data Generation**    For data augmentation methods, we need to generate pseudo training instances from a seed corpus. Considering proposed datasets were mainly collected from dialogues, we pick the OpenSubtitles dataset (Lison and Tiedemann, 2016), which is a large collection of subtitles in various languages for movies and TV programs, as the seed corpus.

### 5.2 ASR Error Correction Model

In multilingual grammatical error correction task, the adoption of multilingual version of pre-trained model has achieved great success (Rothe et al., 2021). So we pick two strong multilingual pre-trained models mT5 and mBART as the code-switching ASR error correction models to verify the effectiveness of our data augmentation method.

Considering that large language models (LLM) have made significant advancements in various NLP tasks. We adopt the most powerful LLM

ChatGPT as the baseline. In our experiment, we test the performance of the GPT-3.5-turbo model on the code-switching ASR error correction task with zero-shot and few-shot settings. The prompt templates for ChatGPT are shown in Appendix F.

### 5.3 Data Augmentation Baseline

There are other ways to generate pseudo code-switching ASR error correction training instances. In our experiment, we implement two data augmentation baselines to compare with our controllable iterative data augmentation method. Firstly, we apply a code-switching text generation approach to get the Chinese-English code-switching texts from a parallel seed corpus. With the code-switching texts, we implement the rule-based and back-translation-based data augmentation methods to improve the ASR error correction models.

**Code-switching Text Generation**    Following previous work (Hussein et al., 2022), we apply the EC theory (Pratapa et al., 2018) to generate Chinese-English code-switching texts. By taking the pairs of parallel sentences in seed corpus as input, the method generates the code-switching texts by replacing the token in the parse tree with its word-level alignments. In our experiment, we adopt the fast-align tool (Dyer et al., 2013) to get the word-level alignments and parse the sentences with a neural graph-based parser(Dozat and Manning).

**Rule-based Method**    The rule-based method is a simple way to generate errors efficiently. We design a rule-based method that generates synthetic training data with five rules: delete, add, replace, shuffle, spelling error. The details of the rules are introduced in Appendix E. With above rules, we can get synthetic training samples with various errors. These synthetic training samples can be used to improve the performance of the code-switching ASR error correction system.

**Back-translation Method**    Inspired by the use of back-translation in downstream tasks (Xie et al., 2018), we train a seq2seq noisy model (like the back-translation model) to generate noisy sentences from correct ones. Then the output of the noising model is paired with the input and the sentence pairs are then used as pseudo data. In our experiment, we consider adopting a base mT5 model as the noising model for our baseline.

| Correction Model | Augmentation Method | Augmented Data Size | SEAME-C | | | | ASCEND-C | | | |
|---|---|---|---|---|---|---|---|---|---|---|
| | | | P↑ | R↑ | $F_{0.5}$↑ | MER↓ | P↑ | R↑ | $F_{0.5}$↑ | MER↓ |
| mT5 | - | - | 26.2 | 18.7 | 24.3 | 21.14 | 20.7 | 15.3 | 19.3 | 40.31 |
| | Rules | 5M | 27.4 | 19.9 | 25.4 | 20.74 | 23.9 | 17.8 | 22.4 | 37.57 |
| | Back-translation | 4.1M | 27.1 | 20.5 | 25.5 | 20.70 | 23.3 | 17.4 | 21.8 | 38.21 |
| | CI | 3.3M | **28.5** | **20.7** | **26.5** | **20.45** | **25.0** | **19.6** | **23.7** | **36.35** |
| mBART | - | - | 23.8 | 17.1 | 22.1 | 21.65 | 19.6 | 14.9 | 18.4 | 40.12 |
| | Rules | 5M | 25.4 | 18.9 | 23.8 | 21.18 | 22.5 | 17.2 | 21.2 | 38.63 |
| | Back-translation | 4.1M | 25.7 | 19.1 | 24.0 | 21.03 | 22.8 | 18.0 | 21.7 | 38.44 |
| | CI | 3.3M | **26.9** | **21.0** | **25.5** | **20.73** | **24.1** | **18.3** | **22.7** | **37.50** |
| GPT-3.5-turbo(zero-shot) | - | - | 7.5 | 8.1 | 7.6 | 40.01 | 5.7 | 3.9 | 5.2 | 71.28 |
| GPT-3.5-turbo(3-shot) | - | - | 13.3 | 12.6 | 13.1 | 35.44 | 7.2 | 4.8 | 6.5 | 65.62 |

Table 5: Comparison results of each method. Augmented data sizes show the amounts of additional training sentences used in each method.

# 6 Results and Analysis

## 6.1 Comparison Results

We evaluate the performance of our controllable iterative data augmentation method on proposed datasets. Table 5 shows the results.

Firstly, we compare our method with the two data augmentation baselines. For the fairness of comparison, we train the correction models with the similar scale of augmentation data. Here we use pseudo data generated from the controllable iterative method with three iterations to train the correction model.

As we can see from the table, our proposed controllable iterative data augmentation method achieves the best performance in both $M^2$ scorer and MER metrics on SEAME-C and ASCEND-C datasets. It outperforms the two baselines for both mT5 and mBART correction methods, which shows the effectiveness and robustness of our controllable iterative method in Chinese-English code-switching ASR error correction task.

Besides, we test ChatGPT (GPT-3.5-turbo) on the proposed dataset. From the table, we can find that the performance of ChatGPT is poor in this task. Under the zero-shot setting, it can only correct a few errors. By learning from examples in the prompt, the performance of ChatGPT under few-shot setting has significantly improved, but still far from the performance of mT5 and mBART models.

The experiment results show that the code-switching ASR error correction task is challenging at present. Due to the scarcity of data and the complexity of code-switching text, LLM method is far from achieving satisfactory results in this task. We present that our data augmentation is an effective method to improve the performance of Chinese-English code-switching ASR error correction models.

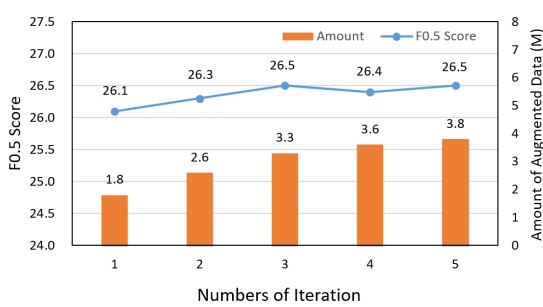

Figure 2: Results of the mT5 correction model on SEAME-C when trained with outputs collected from different iterations of our controllable iterative method.

## 6.2 Iteration of Proposed Method

We build the pseudo training pairs with the outputs of the controllable iterative method in multiple iterations. In this section, we try to explore the influence of the iteration number on the generation of augmented data. Figure 2 shows the results of the mT5 correction model on the SEAME-C dataset when it is trained with outputs collected from different iterations. The details of results are shown in Appendix G.

As the iteration progresses, controllable iterative method continuously generates noisy code-switching data. By collecting the outputs in iterations, we can obtain more pseudo training pairs at each new iteration. In general, using a larger scale of training data may achieve better performance. However, the experimental results show the bottleneck of the data augmentation method that further increasing augmented training data at the 3-th iteration even has a negative impact on performance. This bottleneck might be caused by the quality reduction of the generated data after a few iterations. Considering the balance between perfor-

| Dataset | Augmentation Method | P↑ | R↑ | $F_{0.5}$↑ | MER↓ |
|---|---|---|---|---|---|
| **SEAME-C** | Rules | 27.4 | 19.9 | 25.4 | 20.74 |
| | CI | 28.5 | 20.7 | 26.5 | 20.45 |
| | Rules+CI | **29.2** | **21.3** | **27.2** | **20.15** |
| **ASCEND-C** | Rules | 23.9 | 17.8 | 22.4 | 37.57 |
| | CI | 25.0 | 19.6 | 23.7 | 36.35 |
| | Rules+CI | **25.6** | **20.8** | **24.5** | **35.46** |

Table 6: Results of the mT5 correction model on datasets when trained with the combination of pseudo data generated by controllable iterative (CI) and ruled-based methods.

mance and computing costs, we select the outputs generated in the third iteration as the augmented data used for Chinese-English code-switching ASR error correction task.

### 6.3 Combination of Augmented Data

Rule-based and model-based data augmentation methods are two different approaches to pseudo training data generation. Therefore, the distributions of errors generated by these two types of data augmentation methods might be different in general. In this section, we try to explore the effect of combining the data generated by rule-based and our methods.

In our experiments, we pick the mT5 model that achieves higher evaluation scores in previous experiments as the error correction model. The results of using the combined pseudo data are shown in Table 6. From the results, we can find that training mT5 with the combined pseudo data achieves the best scores on our datasets. The use of the data generated by the rule-based method can further improve the performance of the error correction model.

We further analyze the results of different error types. With the automatic annotator, we evaluate the performance in four types: redundant (R), missing (M), word selection (S), and word ordering (W). We conduct the experiment on our SEAME-C dataset with mT5 as the error correction model. The results are shown in Table 7.

| Error Type | R | M | S | W |
|---|---|---|---|---|
| **Augmentation Method** | $F_{0.5}$ | $F_{0.5}$ | $F_{0.5}$ | $F_{0.5}$ |
| - | 8.1 | 16.3 | 21.1 | 9.8 |
| Rules | 10.8 | **25.6** | 26.1 | 11.1 |
| CI | 17.1 | 20.5 | 28.7 | **14.7** |
| Rules+CI | **18.7** | 21.1 | **29.1** | 7.9 |

Table 7: Results of mT5 correction model on SEAME-C with different augmentation methods on different error types.

As can be seen from the results, data augmentation methods affect the performance of the correction model with respect to different error types. Compared with the controllable iterative method, the rule-based method achieves higher scores on missing errors. The correction model benefits from the operations such as adding and deleting tokens defined in the rules that provide sufficient instances with missing errors. Generally, combining controllable iterative and rule-based data augmentation methods can cover most error types and generate samples with high diversity. It enables the correction model to achieve better performance.

## 7 Conclusion

In this paper, we explore the code-switching ASR error correction task. To enable the research, we first propose two Chinese-English code-switching ASR correction datasets. To address the problem of lacking sufficient training data, we propose the controllable iterative data augmentation method that can generate code-switching ASR error correction instances from the monolingual corpus iteratively. Experiment results show that our method achieves the best performance compared with the rule-based augmentation method, back-translation augmentation method, and ChatGPT method. We further find that combining the data generated by both rule-based method and controllable iterative method can further improve the performance of the error correction model. In future work, we will apply our data augmentation method to other code-switching tasks and also test our method in other language pairs.

## Limitations

In this paper, we only conduct experiments on Chinese-English datasets. There are code-switching phenomena that occur in other languages, for example, the Spanish-English and French-Italian code-switching in America. The empirical conclusions we draw in this study might be changed under different language settings. Another limitation of our study is the lack of manual evaluation. Considering the challenge of the code-switching ASR error correction task, manual evaluation can cover the cases that are evaluated inaccurately by automatic metrics. More interesting conclusions might be drawn by conducting the manual evaluation.

## Acknowledgements

This work was supported by National Key R & D Program of China (2021YFF0901502), National Science Foundation of China (No. 62161160339), State Key Laboratory of Media Convergence Production Technology and Systems and Key Laboratory of Science, Technology and Standard in Press Industry (Key Laboratory of Intelligent Press Media Technology). We appreciate the anonymous reviewers for their helpful comments. Xiaojun Wan is the corresponding author.

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

## A  Processing for Data Construction

### A.1  ASR System

Here is the process we use XLSR-53 model to generate code-switching ASR error correction instances. Firstly, we build the vocabulary from the transcription in the datasets. Based on the vocabulary of 26 English letters, we add all Chinese characters appearing in the transcriptions to the vocabulary for the generation of Chinese. Then, we fine-tune the XLSR-53 model on the training data from SEAME and ASCEND datasets. We pick the parameters with the best performance in the validation set as our code-switching ASR system for next recognition step. With the fine-tuned XLSR-53 model, we can initially obtain the recognition results required for building the code-switching ASR error correction corpus. As the datasets are spontaneous speeches, the corpus is mainly informal and non-speech sounds often occur, such as laughing and coughing. Given that, we design some simple rules to polish the corpus, such as removing non-speech sounds and repeated words.

### A.2  Annotator

The annotator takes three steps: tokenization, alignment, and merging. For tokenization, we use character-level tokenization for Chinese and word-level tokenization for English. Alignment works by computing the alignment score for each pair of tokens in the source and target sentences. In particular, the Cilin thesaurus (Jiaju Mei, 1996) is leveraged to give a similarity score between Chinese characters. Once the alignment score matrix is computed, the sequence with the lowest score is returned. At last, we use a simple merging strategy that merges consecutive sequences of edits of the same type.

## B  Transformations in GECToR

GECToR proposed the token-level transformations for the sequence editing grammatical error correction model. Transformations increase the coverage of grammatical error corrections for limited output vocabulary size for the most common types of errors. The work divided transformations into two types: basic transformations and g-transformations.

Basic transformations perform the most common token-level edit operations, such as: keep the current token unchanged (tag \$KEEP\$), delete current token (tag \$DELETE\$), append new token $t_1$ next to the current token (tag \$APPEND $t_1$ \$) or replace the current token with another token $t_2$ (tag \$REPLACE $t_2$ \$).

G-transformations perform task-specific operations such as: change the case of the current token (\$CASE\$ tags), merge the current token and the next token into a single one (\$MERGE\$ tags) and split the current token into two new tokens (\$SPLIT\$ tags). Moreover, tags from \$NOUN NUMBER\$ and \$VERB FORM\$ transformations encode grammatical properties for tokens.

## C  Processing for Sequence Editing Model

### C.1  Processing for Transformations

It takes a three-step pre-processing algorithm to convert a sentence to a sequence of transformations. Firstly, it maps each token from source sentence to sub-sequence of tokens from target sentence. For this purpose, it detects the minimal spans of tokens which define differences between source tokens and target tokens. Thus, such a span is a pair of selected source tokens and corresponding target tokens. To get tags on the token level, it searches for the best-fitting sub-sequence of target tokens by minimizing the modified Levenshtein distance for each source token. Secondly, for each mapping in the list, it finds token-level transformations which convert source token to the target sub-sequence. Finally, it leaves only one transformation for each source token. Through the above three steps, we map the tag to each single source token.

## C.2 Processing for Translation

Our model is aimed to generate code-switching sentences with errors from correct monolingual sentences, but the training pairs in datasets only consist of code-switching texts. Considering that, we adopt an open translation tool OPUS-MT (Tiedemann and Thottingal, 2020) to translate the code-switching ASR transcriptions into monolingual sentences. Specifically, we translate the spans of English texts to Chinese and get the Chinese sentences. With regarding the translated monolingual sentence as source and the code-switching sentence with errors from error correction datasets as target, we train the sequence editing model.

## D Implementation Details

In our work, we adopt PyTorch (Paszke et al., 2019) framework and HuggingFace (Wolf et al., 2020) tool to implement all deep neural models. Each model is trained on four NVIDIA A40 GPUs. The details of each model are described as follows.

### D.1 ASR Error Correction Model

Considering that mT5 and mBART have achieved strong performance in many multilingual tasks, we adopt these pre-trained models to correct ASR errors. In the experiments, we use the base version of mT5 and fine-tune it by using the Adam optimization method with learning rate 5e-4, weight decay parameter 0.01, dropout rate 0.1, and batch size 24. Besides, we use the large version of mBART, and fine-tune it by using the Adam optimization method with learning rate 2e-5, dropout rate 0.1, and batch size 16. Beam search with beam size of 5 is used for decoding.

### D.2 Sequence Editing Model

The sequence editing model is an encoder made up of Transformer architecture stacked with two linear layers with softmax layers. We adopt the base version of m-BERT to initialize the encoder. Our model consists of 12 Transformer layers and two linear layers. We train the model with learning rate 1e-5, weight decay parameter 0.01, dropout rate 0, and batch size 32. The size of the vocabulary is 12229. It consists of 4971 basic transformations, 29 token-independent g-transformations and 7229 cross-lingual transformations.

### D.3 Pseudo Data Generation Algorithm

We add a confidence bias to control the decoding process. In our experiments, the confidence bias $\epsilon_0$ is to add -1.0 to the generation probability of all transformations except \$KEEP\$, \$TRANS-LATION\$ and other cross-lingual transformations. The confidence bias $\epsilon_1$ is set to keep the original generation probability of all transformations. The threshold $\lambda$ is set to 0.85.

| Method | Prompt Template |
|---|---|
| Zero-shot | "role": "system", "content": "You are a Chinese-English code-switching ASR error correction tool that can identify and correct errors in the text. Especially, code-switching is a linguistic phenomenon where a speaker alternates between two or more languages or language varieties within a single conversation or even within a single sentence." "role": "user", "content": "Please identify and correct any errors in the following sentence indicated by <input> ERROR </input> tag, you need to comprehend the sentence as a whole before gradually identifying and correcting any errors while keeping the original sentence structure unchanged as much as possible. Remember to format your corrected output results with the tag <output> Your Corrected Version </output>. Please start: <input> Input Sentence </input>:" |
| Few-shot | "role": "system", "content": "You are a Chinese-English code-switching ASR error correction tool that can identify and correct errors in the text. Especially, code-switching is a linguistic phenomenon where a speaker alternates between two or more languages or language varieties within a single conversation or even within a single sentence." "role": "user", "content": "Please identify and correct any errors in the following sentence indicated by <input> ERROR </input> tag, you need to comprehend the sentence as a whole before gradually identifying and correcting any errors while keeping the original sentence structure unchanged as much as possible.Here are some in-context examples:
(1), <input> SRC-1 </input>: <output> TGT-1 </output>;
(2), <input> SRC-1 </input>: <output> TGT-1 </output>;
...
(n), <input> SRC-n </input>: <output> TGT-n </output>;
Please feel free to refer to these examples. Remember to format your corrected output results with the tag <output> Your Corrected Version </output>. Please start: <input> Input Sentence </input>:" |

Table 8: Zero-shot and few-shot prompt template for Chinese-English code-switching ASR error correction.

# E    Rule-based Method

The details of the five rules adopted in our proposed rule-based method are introduced in the following.

**Delete.** Randomly delete a token with a probability.

**Add.** Firstly, randomly select a word or Chinese character from the vocabulary of the ASR datasets, and then add the selected word or character to a random position.

**Replace.** Randomly replace a token with a word or Chinese character from the vocabulary of the ASR datasets. In particular, if the selected token is a Chinese character, we select the replacement from the homophones with a probability.

**Shuffle.** Shuffle the tokens by adding a normal distribution bias to the positions of the tokens.

**Spell Error.** Randomly apply spell error to an English word. We randomly perturb letters with operations, i.e. substitution, deletion, insertion or transposition of letters.

# F    Prompt Template

Zero-shot and few-shot prompt templates for Chinese-English code-switching ASR error correction are shown in Table 8.

# G    Additional Experiment Results

Results of the mT5 correction model on SEAME-C, when trained with outputs collected from different iterations of our method, are shown in Table 9.

| Iteration | Augmented Data Size | P↑ | R↑ | $F_{0.5}$↑ | MER↓ |
|---|---|---|---|---|---|
| Iteration 1 | 1.8M | 28.0 | 20.3 | 26.1 | 20.54 |
| Iteration 2 | 2.6M | 28.2 | 20.6 | 26.3 | 20.49 |
| Iteration 3 | 3.3M | 28.5 | 20.7 | 26.5 | 20.45 |
| Iteration 4 | 3.6M | 28.3 | 20.8 | 26.4 | 20.51 |
| Iteration 5 | 3.8M | 28.3 | 21.1 | 26.5 | 20.40 |

Table 9: Results of the mT5 correction model on SEAME-C when trained with outputs collected from different iterations of our method.