# OpenReview forum: "New Datasets and Controllable Iterative Data Augmentation Method for Code-switching ASR Error Correction"
_EMNLP/2023/Conference — EMNLP 2023 Findings_

### Official Review · Reviewer_XFMs · 2023-08-04

**Soundness:** 4

**Excitement:**

3: Ambivalent: It has merits (e.g., it reports state-of-the-art results, the idea is nice), but there are key weaknesses (e.g., it describes incremental work), and it can significantly benefit from another round of revision. However, I won't object to accepting it if my co-reviewers champion it.

**Missing References:**

N/A

**Paper Topic And Main Contributions:**

The paper is about datasets and approaches for code-switching ASR error correction. It investigates ASR error correction in the context of code-switching for the first time. It creates a new set of training and evaluation datasets (SEAME-C and ASCEND-C) using pairs of ground truth transcripts from SEAME/ASCEND (code-switching ASR dataset) and ASR hypotheses from a finetuned multilingual ASR model (XLSR-53). It also proposes a new data augmentation strategy to cheaply generate synthetic but noisy code-switching ASR error correction examples from large monolingual datasets and shows some improvement over baselines on the newly proposed datasets.

**Questions For The Authors:**

A. Is WER the right metric to evaluate for English-Chinese code-switching? I believe past work (e.g. https://arxiv.org/pdf/1810.13091.pdf ) uses Mixed Error Rate (MER) which computes word-level errors for English and character-level errors for Chinese. Adding MERs would make the results more convincing.

B. How do you finetune the mBERT model on your predefined transformations; if you have predefined deterministic transformations (APPEND, REPLACE, etc.) what is the purpose of using an mBERT model to generate these same transformations?

C. What is the input and output format of your mBERT model? What exactly is in the 12229 sized output vocabulary?

D. What method do you use to do controllable decoding from the mBERT model?

E. What dataset do you use to train the mBERT sequence editing model? Appendix C explains that you use OPUS-MT to translate code-switched data to monolingual data, and use this to train; where does this code-switched data come from?

F. On lines 160-161 you mention that you remove monolingual sentences from the val/test sets. Why not do the same for the train sets as well? Relatedly, are the code-mixing statistics in Table 3 on the training set (since you remove monolingual sentences from Val/test, the proportion of Chinese/English sentences should be 0.00% for those)? Please clarify in the text.

G. “This type of error is caused by mistakes ... with similar pronunciation” on lines 243-245 ; was this verified with an experiment? Word selection errors could be due to replacements with unrelated words as well. It would be good to support the statement with an experiment.

H. Can you motivate the choice of XLSR-53 as the base ASR model as well as why you opted to finetune it on the code-switching datasets in order to generate synthetic ASR hypotheses? Why not use a) models from previous papers trained explicitly for code-switching e.g. the SEAME model from ESPNet or b) try zero-shot non-finetuned models like the base XLSR-53 model or Whisper?

I. What ASR model was used in Table 5. What is the WER of this ASR model without any error-correction applied? (Adding that WER to Table 5 will be helpful).

J. The inability of an LLM to 3-shot adapt to a novel task does not imply the task is fundamentally challenging (Lines 499-504); is there more evidence for your statement?

**Reasons To Accept:**

1. This paper is a very good investigation into the task of ASR error correction in the code-switched setting which has been previously unexplored and hence is a very novel investigation.
2. The authors perform insightful data analysis in Section 3.2 and use it to guide their approach to generate synthetic ASR error correction data.
3. The choice of baselines is good and covers a variety of plausible approaches for data augmentation.
4. I especially liked Sections 6.2 and 6.3 that provide very interesting insights on how many iterations matter and what types of errors are fixed by what kinds of data augmentation methods.
5. The paper says that the code (new approach) and data (2 new datasets) will be released, which is quite beneficial to the community.

**Reasons To Reject:**

1. The data used to train baselines compared against in Section 5.3 are not properly explained; although the authors mention that they control for dataset size, it is hard to verify whether they are fair comparisons or not.
2. The training and evaluation datasets in this work (SEAME-C and ASCEND-C) are created by finetuning XLSR-53 on SEAME and ASCEND. All ASR correction models in this work are trained and evaluated on SEAME-C and ASCEND-C, and based on my reading, the XLSR-53 ASR model is also used in the evaluation phase to report M^2 and WER metrics in Table 5. As such, the setup seems to have been fit closely to the XLSR-53 ASR model, SEAME and ASCEND. It is not clear from the paper whether the results are generalizable outside of this setting; e.g., can the ASR correction model trained by the authors correct ASR errors from models other than XLSR-53 (or is SEAME-C and ASCEND-C too sensitive to XLSR-53 errors) , and on evaluation datasets other than SEAME-C and ASCEND-C. This may limit the applicability of the proposed datasets and method. For example, it will be good if the authors can add more rows to Table 5 applying their correction model on other code-switching ASR models apart from XLSR-53.
3. Many important details of the method used (Data Construction in Sec 3.1 and the Data Augmentation method in Section 4) are not explained properly, which currently affects the quality and reproducibility of the paper. I suggest
    1. moving some info (e.g. the data used to finetune XLSR-53, a summary of the transformations used, etc.) from the Appendix to the main text.
    2. adding answers to my questions (in ‘Questions for the Authors’) to complete the missing pieces from the description. Answering these questions is likely to increase my Soundness score.

**Reproducibility:**

3: Could reproduce the results with some difficulty. The settings of parameters are underspecified or subjectively determined; the training/evaluation data are not widely available.

**Reviewer Confidence:**

4: Quite sure. I tried to check the important points carefully. It's unlikely, though conceivable, that I missed something that should affect my ratings.

**Typos Grammar Style And Presentation Improvements:**

- Line 117: swinging -> switching
- Add citations to lines 034 and 417
- Add word-level English translations to Figure 1 to help non-Chinese-speaking readers to understand the code-mixing better
- Specify the version of ChatGPT / gpt-3.5-turbo used (e.g. 0314 or the live version at chat.openai.com) and the hyperparams used (temperature etc.) if any.
- There is no description about the precision, recall and F1 evaluation metrics (from the MaxMatch scorer) in Table 5; adding a description would be important.

---

> ### Author Rebuttal · Authors · 2023-08-29
>
> Thanks for your careful and valuable reviews. We appreciate your time and effort in reviewing our work. The improvement suggestions you provide are beneficial, and we will modify our paper based on them. Besides, we provide detailed responses to your questions in the following sections, hoping to address your concerns.
>
> ### Responses for Reasons to Reject
>
> **Q1**: Ambiguity of the data used to train baselines.
>
> **A1**: Certainly, data is the key to the methods. In our experiment, the rule-based method, back-translation method, and the proposed CI method adopt the same 5 million sentences sampled from the parallel seed corpus. In particular, the CI method and the back-translation method use the similarity discriminator to filter sentence pairs with low similarity, resulting in synthetic sentences of fewer than 5 million. Since the sources used for data augmentation are identical, the comparison of the methods is entirely fair and reliable.
>
> **Q2**: The limitations of using the single ASR system XLSR-53.
>
> **A2**: XLSR-53 is the multilingual version of wav2vec 2.0 model pre-trained on 53 languages, which shows strong performance and robustness in the code-switching ASR task. It has been widely employed as a strong baseline in previous works [1,2].
>
> Besides, we evaluate the performance of the XLSR-53 and another ASR system ETEH [3]  before constructing the datasets. The results are shown as follows.
>
> | Dataset | Method | MER |
> | --- | --- | --- |
> | SEAME | XLSR-53 | 24.02 |
> | SEAME | ETEH |31.62 |
> | ASCEND | XLSR-53 | 42.38|
> | ASCEND| ETEH |44.85 |
>
> In particular, we replace the term WER with MER (Mixture Error Rate) which is a more accurate enunciate. It can be observed that the performance of the XLSR-53 system on both datasets is better than the ETEH system. Considering the above reasons, we adopt XLSR-53 ASR system for constructing the datasets, to ensure their quality and generality. For sure, we acknowledge the utilization of a single system inherently carries limitations. We will make sure to supplement our experiment with other typical ASR systems (e.g. Whisper, ESPNet) in the revised version of the paper.
>
> **Q3**: Many important details of the method used are not explained properly.
>
> **A3**: Due to the limitation of pages, we select to elaborate on the essential aspects of the main text. In the revised version, we will carefully consider your suggestions and move some content from the appendix to the main body.
>
> ### Responses for Questions
>
> The questions you raised are valuable and helpful to the refinement of our paper. Due to the limitation of pages, some details are not explained properly. In this section, we provide detailed responses to your questions, hoping to complete the missing pieces of the description and address your concerns.
>
> **Q1**: Is WER the right metric to evaluate for English-Chinese code-switching?
>
> **A1**: The question you raise is highly helpful. We have referred to preview work [4] that uses WER as the metric, which actually calculates the error rate of Chinese characters and English words as same as the MER. After reading the paper you mentioned in question [5], we searched for related works [6] and found that using the term Mixture Error Rate (MER) is more common. In the revised version, we will replace the imprecise term WER with MER.
>
> **Q2**: The fine-tuning process of the mBERT model.
>
> **A2**: As mentioned in section 4.1.3, we train the sequence editing model by the sequences of transformations that are converted from the source sentences. The fine-tuning process of mBERT model is conducted as a component of the overall training process. It encodes the source sentence and outputs feature representations to the following linear layers.
>
> **Q3**: The purpose of using the mBERT model.
>
> **A3**: We refer to the previous GECToR method, which adopts pre-trained transformers as the encoders and conducts experiments with several pre-trained settings. Their experiments show that utilizing pre-trained models can accelerate the convergence speed of training and improve the performance of method.
>
> Considering that, we apply the pre-trained models to our method. We conduct the experiments with several multilingual pre-trained models (e.g. mBERT, mBART, mT5) and find that the mBERT model achieves the best results. So we select it for our method to accelerate the convergence process in training and improve the robustness and performance of our method.
>
> **Q4**: What is the input and output format of your mBERT model?
>
> **A4**: As mentioned in A2, the inputs of the mBERT are the source sentences and the outputs are the feature presentations with dim matching the hidden size of mBERT. In particular, the model adopts WordPiece tokenizer to process the inputs.
>
> **Q5**: What exactly is in the 12229 sized output vocabulary?
>
> **A5**: As mentioned in appendix D.2, the vocabulary consists of 4971 basic transformations,
> 29 token-independent g-transformations and 7229 cross-lingual transformations. The basic transformations perform the most common token-level edit operations, such as: KEEP, DELETE, APPEND. The g-transformations perform task-specific operations such as: CASE,  MERGE. The cross-lingual transformations perform the translation operation. The Appendix B shows the details.
>
> **Q6**: What method do you use to do controllable decoding from the mBERT model?
>
> **A6**: As mentioned in section 4.1.2, we control the decoding process by adjusting the confidence bias which is a parameter applied to change the generation probability of transformations. In our method, we get the generation probability of each transformation by the linear layers with softmax layers. Then we apply the confidence bias which is the vector composed of float numbers to the generation probability. For example, adding a positive number to KEEP can increase the probability of generating the KEEP. By adjusting the confidence bias, the decoding process of sequence editing model is under control.
>
> **Q7**: The dataset used to train the mBERT sequence editing model.
>
> **A7**: As mentioned in appendix C, we train the mBERT sequence editing model by the data that is translated from code-switching sentences. In particular, we select the code-switching sentences from the training sets of ASCEND-C and SEAME-C for translation.
>
> **Q8**: Why remove monolingual sentences for the val/test sets?
>
> **A8**: The datasets we proposed focus on errors in Chinese-English code-switching sentences. However, the original datasets contain monolingual sentences. So, we remove the monolingual sentences for the val/test sets to ensure that the evaluation is performed in code-switching sentences. Besides, in order to provide more sufficient training data, we make these monolingual data available as well. Users have the option to decide whether to adopt these monolingual sentences to their training process.
>
> In addition, the proportions of Chinese/English sentences refer to the ratio across the entire dataset, including the training, validation, testing sets. Therefore, the proportion is non-zero and accurate.
>
> **Q9**: The verification of the statements that the word selection errors are caused by mistakes in the recognition of words or phrases with similar pronunciation.
>
> **A9**: Thanks for pointing out the issues. By observing the cases, we find that the majority of errors are caused by mistakes in the recognition of words or phrases with similar pronunciation.
> However, the concerns you raised that word selection errors could be due to replacements with unrelated words do exist. In the revised version, we will revise the statement as “This type of error is mainly caused by mistakes in the recognition of words or phrases with similar pronunciation.”
>
> **Q10**: The choice of ASR system.
>
> **A10**: As mentioned in A2 of responses for reasons to reject part, XLSR-53 is a method with strong performance and robustness in the code-switching ASR task, and it has been widely employed as a strong baseline in previous works. So, we adopt it to build the datasets with quality. We conduct tests on XLSR-53 and ETEH models in previous experiments. We will make sure to supplement our experiment with other typical ASR systems in the revised version of the paper. As for the use of zero-shot ASR model, they achieve poor performance (WER > 70%) in our experiments, so we do not consider them.
>
> **Q11**: The details of the results in Table 5.
>
> **A11**: As mentioned in Section 3, the results are on the dataset constructed by XLSR-53 model. So the ASR model is XLSR-53. The WERs of it without any error-correction applied are shown in A2 of responses for reasons to reject part.
>
> **Q12**: More evidence for the statement that the task is fundamentally challenging.
>
> **A12**: GPT 3.5 has achieved great success in many NLP tasks. However, its performance in the Chinese-English code-switching ASR error correction task is far from satisfactory. Besides, the pre-trained models, such as mT5, mBART, that show effectiveness in error correction tasks in the previous works do not achieve a great performance on this task. So, we state that the Chinese-English code-switching ASR error correction task is quite challenging under the existing techniques.
>
> Indeed, the issue you pointed out has some rationality. We will make sure to supplement further experiments with various LLMs to provide substantiating evidence.
>
> Thank you again for your careful and valuable comments and guidance.
>
> ### References
>
> [1] Ascend: A spontaneous chinese-english dataset for code-switching in multi-turn conversation. Lovenia H, Cahyawijaya S, Winata G I, et al. (2021).
>
> [2] TEVR: Improving Speech Recognition by Token Entropy Variance Reduction. Krabbenhöft, H. N., & Barth, E. (2022).
>
> [3] Eteh: Unified attention-based end-to-end asr and kws architecture. Cheng G, Miao H, Yang R, et al. (2022).
>
> [4] Code-Switching in Automatic Speech Recognition: The Issues and Future Directions. Mustafa M B, Yusoof M A, Khalaf H K, et al. (2022).
>
> [5] Towards end-to-end code-switching speech recognition. Luo N, Jiang D, Zhao S, et al. (2018).
>
> [6] Summary On The ISCSLP 2022 Chinese-English Code-Switching ASR Challenge.Deng S, Li C, Bai J, et al. (2022).

---

### Official Review · Reviewer_BZT9 · 2023-08-05

**Soundness:** 3

**Excitement:**

3: Ambivalent: It has merits (e.g., it reports state-of-the-art results, the idea is nice), but there are key weaknesses (e.g., it describes incremental work), and it can significantly benefit from another round of revision. However, I won't object to accepting it if my co-reviewers champion it.

**Paper Topic And Main Contributions:**

Authors propose new approach to improving the performance of automatic speech recognition (ASR) error correction systems in code-switching settings (more than one language used in the same conversation). For the task they propose and develop 2 Chinese-English code-switching datasets. They then propose a Controllable Iterative (CI) data augmentation method that combines rule-based and machine learning-based approaches to generate synthetic training data for ASR error correction models. They evaluate the proposed method on the dataset developed for the task. They end with possibility of new research areas in future.

**Reasons To Accept:**

1.	Well structured, detailed and crisp paper.
2.	Proposes a new corpus of dataset for the task.
3.	Proposes a new approach to improving ASR error correction in code-switching settings.
4.	Proposes a Controllable Iterative (CI)  data augmentation method that combines rule-based and machine learning-based approaches.
5.	Provides detailed descriptions of the datasets and experimental setup.


**Reasons To Reject:**

1.	Clarity on adding assumptions across the board.
2.	Point of view on Generalizability.
3.	Manual annotation and evaluation by humans can further improve the system.
4.	Any kind of ablation study can help understanding the modularized aspect of the system and can open up further research areas.
5.	Details/explanation around the configuration/hyperparameters of the rule-based and machine learning-based approaches used can help with more clarity.
6.	Paper can benefit further from having detailed analysis of the computational complexity or efficiency of the proposed method.


**Reproducibility:**

3: Could reproduce the results with some difficulty. The settings of parameters are underspecified or subjectively determined; the training/evaluation data are not widely available.

**Reviewer Confidence:**

3: Pretty sure, but there's a chance I missed something. Although I have a good feel for this area in general, I did not carefully check the paper's details, e.g., the math, experimental design, or novelty.

---

> ### Author Rebuttal · Authors · 2023-08-29
>
> Thank you for reviewing our paper and providing us with valuable comments. We appreciate your time and effort in reviewing our work. In response to your comments, we would like to address the concerns raised:
>
> **Q1**: Clarity on adding assumptions across the board.
>
> **A1**: In this paper, we present a data augmentation approach to assist in the code-switching ASR error correction task. Our method is based on the assumption that a greater amount of data can achieve better performance. The efficacy of data augmentation methods has been widely demonstrated in previous works [1, 2, 3]. In this work, we further substantiate it through experiments.
>
> **Q2**: Point of view on generalizability.
>
> **A2**: The lack of data is a common challenge for the research in cross-lingual and low-resource languages. Our proposed CI data augmentation method presents an approach to address the data scarcity issue for Chinese-English code-switching ASR error correction tasks. We anticipate that our method can be extended to relevant code-switching tasks involving other languages in the future. Furthermore, our data augmentation methodology provides insights to researchers studying other tasks in cross-lingual and low-resource languages.
>
> **Q3**: Manual annotation and evaluation by humans can further improve the system.
>
> **A3**: Thanks for your suggestions. We will incorporate manual annotation and human evaluation in our work to further improve the quality of the datasets and the evaluation results.
>
> **Q4**: Any kind of ablation study can help in understanding the modularized aspect of the system and can open up further research areas.
>
> **A4**: In section 6, we present the related contents, hoping to address your concerns.
>
> In section 6.2, we try to explore the influence of the iteration number on the generation of augmented data. In general, using a larger scale of training data may achieve better performance. However, the experimental results show the bottleneck of the data augmentation method.
>
> Data plays a crucial role in our proposed method. In section 6.3, we try to explore the effect
> of combining the data generated by different methods. From the results, we can find that the use of the data generated by the rule-based method can further improve the performance of the error correction model.
>
> **Q5**: Details/explanation around the configuration/hyper-parameters of the rule-based and machine learning-based approaches used can help with more clarity.
>
> **A5**: In Appendix E, we provide a detailed explanation of the rule-based approach. In particular, the error probabilities for the operations "delete," "add," "replace," "shuffle," and "spell error" are 0.1, 0.1, 0.5, 0.1, and 0.2, respectively. As for the machine learning-based method, we adopt a base mT5 model and fine-tune it by using the Adam optimization method with learning rate 2e-4, weight decay parameter 0.01, dropout rate 0.1, and batch size 24. Further details of the configuration and hyper-parameters can be referred to the released code.
>
> **Q6**: Paper can benefit further from having detailed analysis of the computational complexity or efficiency of the proposed method.
>
> **A6**:
> In this work, we proposed a data augmentation method to improve the performance of Chinese-English code-switching ASR error correction model by generating synthetic training data. The method only affects the training data of the error correction model and does not impact the speed of the inference process.
>
> Thank you again for your careful and valuable comments and guidance.

---

### Official Review · Reviewer_6CD8 · 2023-08-05

**Soundness:** 3

**Excitement:**

3: Ambivalent: It has merits (e.g., it reports state-of-the-art results, the idea is nice), but there are key weaknesses (e.g., it describes incremental work), and it can significantly benefit from another round of revision. However, I won't object to accepting it if my co-reviewers champion it.

**Paper Topic And Main Contributions:**

The paper presents the ASR error correction task for Chinese-English code-switching dialogues. The authors created new datasets with ASR outputs and automatic error type annotations from existing ASR corpora SEAME and ASCEND. They further introduced a controllable iterative (CI) data augmentation method to amplify the training data for error correction, and showed convincing improvements over other data augmentation techniques.

**Questions For The Authors:**

- Please report the WER scores on the ASR outputs before error correction is applied.
- The translation transformation replaces the current token with the most common translation in a Chinese-English dictionary (line 300). Is the (subword?) tokenization of the sequence editing model compatible with the dictionary?
- Will the generated synthetic training data also be published?

**Reasons To Accept:**

- The proposed CI data augmentation process is well motivated by the error analysis of the datasets they introduced, and carefully designed targeting the ASR error correction goal.
- They provide comprehensive details on dataset creation and pseudo data generation.
- They did not only conduct the evaluation in M^2 / WER, but also analyzed error types and the influence of iteration numbers, and even included the attempts with LLM prompting, which supports deeper understanding of the task.

**Reasons To Reject:**

A discussion of the error statistics across different ASR systems would be missing. Their ASR error analysis relies on the outputs from a particular ASR system only. Moreover, their ASR model choice might be questionable especially with the focus on the dialogue in Chinese-English in particular. ASCEND paper [1] reports that the wav2vec model achieved 35.30 WER, while they report 40.31 WER for the baseline error correction model without additional augmented data.

[1] Lovenia et. al., ASCEND: A Spontaneous Chinese-English Dataset for Code-switching in Multi-turn Conversation, LREC 2022.

**Reproducibility:**

3: Could reproduce the results with some difficulty. The settings of parameters are underspecified or subjectively determined; the training/evaluation data are not widely available.

**Reviewer Confidence:**

3: Pretty sure, but there's a chance I missed something. Although I have a good feel for this area in general, I did not carefully check the paper's details, e.g., the math, experimental design, or novelty.

**Typos Grammar Style And Presentation Improvements:**

- line 976: [...] are shown in **Table** 8.
- line 979: [...] are shown in **Table** 9.
- Some transformation operation names are surrounded by **$** (line 294, line 365), others don't have the ending **$**.
- To demonstrate how well the generated pseudo data can mimic the true code-switching target and source with ASR errors, it would be nice to have some real examples found in the corpus. For instance, is there any comparable counterpart of the sample generation in Figure 1 ("我在 Singapore make 很多 good friends") that represents a typical code-switching in an actual dialogue?

---

> ### Author Rebuttal · Authors · 2023-08-29
>
> Thanks for your interest in our work. We appreciate your valuable comments and suggestions for improvement. In response to your questions, we would like to address each of the concerns raised.
>
> ### Responses for Questions
>
> **Q1**: A discussion of the error statistics across different ASR systems would be missing.
>
> **A1**: XLSR-53 is the multilingual version of wav2vec 2.0 model pre-trained on 53 languages, which shows strong performance and robustness in the code-switching ASR task. It has been widely employed as a strong baseline in previous works [1,2].
>
> Besides, we evaluate the performance of the XLSR-53 and another ASR system ETEH [3]  before constructing the datasets. The results are shown as follows.
>
> | Dataset | Method | MER |
> | --- | --- | --- |
> | SEAME | XLSR-53 | 24.02 |
> | SEAME | ETEH |31.62 |
> | ASCEND | XLSR-53 | 42.38|
> | ASCEND| ETEH |44.85 |
>
> In particular, we replace the term WER with MER (Mixture Error Rate) which is a more accurate enunciate. It can be observed that the performance of the XLSR-53 system on both datasets is better than the ETEH system. Considering the above reasons, we adopt XLSR-53 ASR system for constructing the datasets, to ensure their quality and generality. For sure, we acknowledge the utilization of a single system inherently carries limitations. We will make sure to supplement our experiment with other typical ASR systems (e.g. Whisper, ESPNet) in the revised version of the paper.
>
> **Q2**: The results of the ASCEND paper are better than the results of baseline ASR system in our paper.
>
> **A2**: We try to reproduce the method in the ASCEND paper, but we do not achieve the same level of performance. As mentioned in A1, the XLSR-53 method we select is strong and representative. So we adopt it to build the datasets.
>
> **Q3**: The WER scores on the ASR outputs before error correction is applied.
>
> **A3**: As mentioned in A1, the table shows the scores on the ASR outputs before error correction is applied. The scores for SEAME and ASCEND are 24.02 and 42.38 respectively.
>
> **Q4**: Is the tokenization of the sequence editing model compatible with the dictionary?
>
> **A4**: To process the transformation at the token-level, we take the first subword per token
> from the encoder’s representation to subsequent linear layers. With the process, the tokenized source sentence can be compatible with the dictionary at the same token level.
>
> **Q5**: Will the generated synthetic training data also be published?
>
> **A5**: Of course, we will release the generated synthetic training data along with the released code and data. We hope that the released resources can assist researchers in further investigating this task.
>
> Thank you again for your careful and valuable comments and guidance.
>
> ### References
>
> [1] Ascend: A spontaneous chinese-english dataset for code-switching in multi-turn conversation. Lovenia H, Cahyawijaya S, Winata G I, et al. (2021).
>
> [2]TEVR: Improving Speech Recognition by Token Entropy Variance Reduction. Krabbenhöft, H. N., & Barth, E. (2022).
>
> [3]Eteh: Unified attention-based end-to-end asr and kws architecture. Cheng G, Miao H, Yang R, et al. (2022).

---

### Meta-Review · Area_Chair_8jSu · 2023-09-22

**Recommendation:** 3

**Metareview:**

This is an interesting work that addresses for the first time ASR error correction in code-switched data (Chinese-English). The paper proposes data augmentation strategies for training ASR error correction models. The ASR error correction datasets are first created using pairs of outputs from an ASR model (XLSR-53) and ground truth transcripts, and automatically assigning error types. Then the paper proposes an approach to data augmentation via a set of text transformations.

The paper should include more detail on the dataset construction and the data augmentation approach. As the authors use a single ASR model to construct the datasets and to analyze the resulting errors, it is unclear how the choice of the model would affect to constructed dataset and the results on the error correction task. The paper can be strengthened by experiments on a different ASR model.
The related work section should  include research on the more general error correction task and error correction on monolingual ASR (including specific examples would help the reader). How can insights from these tasks benefit the ASR error correction? Can you use existing data augmentation techniques from the general error correction task for ASR error correction?

---

### Decision · Program_Chairs · 2023-10-07

**Decision:**

Accept-Findings

**Comment:**

This is an interesting work that addresses for the first time ASR error correction in code-switched data (Chinese-English). The paper proposes data augmentation strategies for training ASR error correction models. The ASR error correction datasets are first created using pairs of outputs from an ASR model (XLSR-53) and ground truth transcripts, and automatically assigning error types. Then the paper proposes an approach to data augmentation via a set of text transformations.

The paper should include more detail on the dataset construction and the data augmentation approach. As the authors use a single ASR model to construct the datasets and to analyze the resulting errors, it is unclear how the choice of the model would affect to constructed dataset and the results on the error correction task. The paper can be strengthened by experiments on a different ASR model.
The related work section should  include research on the more general error correction task and error correction on monolingual ASR (including specific examples would help the reader). How can insights from these tasks benefit the ASR error correction? Can you use existing data augmentation techniques from the general error correction task for ASR error correction?